

# The Andean Ibis (*Theristicus branickii*) in South America: potential distribution, presence in protected areas and anthropic threats

Nivia Luzuriaga-Neira[1], Keenan Ennis[2], Michaël A.J. Moens[3], Jose Leon[3], Nathaly Reyes[1], Agusto Luzuriaga-Neira[4,5], Jaime R. Rau[6] and Roxana Rojas-VeraPinto[7]

[1] Unidad de Estudios de la Vida Silvestre-Facultad de Medicina Veterinaria, Universidad Central del Ecuador, Quito, Pichincha, Ecuador
[2] School of Natural Resources, Knoxville College, Knoxville, TN, United States of America
[3] Fundación de Conservación Jocotoco, Quito, Pichincha, Ecuador
[4] Department of Ornithology, American Museum of Natural History, New York, NY, United States of America
[5] Biology Department, University of Nevada - Reno, Reno, NV, United States of America
[6] Laboratorio de Ecología, Departamento de Ciencias Biológicas & Biodiversidad, Universidad de los Lagos, Osorno, Chile
[7] Proyecto Isnachi, La Libertad, Lima, Peru

Corresponding author
Nivia Luzuriaga-Neira,
nluzuriaga@uce.edu.ec

## ABSTRACT

The avifauna of South America is one of the most widely studied groups of vertebrates. However, certain species, such as the Andean Ibis (*Theristicus branickii*), have received limited attention regarding their ecological patterns, biology, current distribution, and environmental requirements. This study analyzed observation data from the Global Biodiversity Information Facility (GBIF) on the Andean Ibis in four countries to identify and understand critical variables that determine the species' presence, assess the proportion of its habitat within protected areas and identify possible threats to the species. Additionally, this study considered environmental and ecological variables to model ecological niches using the maximum entropy approach in MaxEnt to map the suitable habitat of the species. The findings revealed the extent of suitable Andean Ibis habitats in Ecuador, Peru, Bolivia and Chile. The variables that most determined the presence of the species were: altitude (36.57%), distance to lakes (23.29%) and ecological isothermality (13.34%). The distribution area of the Andean Ibis totaled 300,095.00 km$^2$, spanning both sides of the Andean mountains range. Human activities have left a significant impact on the Andean Ibis habitat, with 48% of this area impacted by the human footprint and only 10% of the territory falling within protected areas designated by the respective countries. The results of this study show that the Andean Ibis presents characteristics of a specialist species due to its adaptation to the climate conditions of the plateau and highlands, including low temperatures, herbaceous vegetation and the presence of water bodies. The species is distributed in disconnected Andean landscape areas, whose functionality could be compromised by increased human activities. Complementary studies will be necessary to understand the ecological role and effectiveness of protected areas for conserving the species.

## INTRODUCTION

The tropical Andes avifauna is a biological group that has been extensively studied (*Vuilleumier, 1969*; *Vuilleumier, 1984*; *Vuilleumier & Simberloff, 1980*; *Fjeldså, 1988*; *Valencia & Franke, 1989*), especially their ability to adaptat and evolve across the bioregions they inhabit (*Hazzi et al., 2018*). In high mountain ecosystems above 2,500 m a.s.l. (*Sevillano-Ríos, Rodewald & Morales, 2020*), birds have evolved adaptive strategies to cope with extreme environmental conditions (*DuBay & Witt, 2014*; *Sevillano-Ríos, Rodewald & Morales, 2020*; *Anthelme & Peyre, 2020*). As a result, this taxa has a high rate of endemism and habitat specialization (*Long, Crosby & Stattersfield, 1996*). Although high elevation range Andean birds have been extensively studied, there is limited ecological information for some taxons, especially in relation to potential human threats and the effectiveness of protected areas in conserving their habitats (*Bax & Francesconi, 2019*).

High mountain ecosystems of the tropical Andes are highly fragile and susceptible to the adverse effects of climate change and land use alterations (*Anthelme & Peyre, 2020*), which negatively impact the distribution patterns and abundance of functional groups of birds (*Cardenas et al., 2022*), plants (*Carilla et al., 2023*) and insects (*Moret et al., 2016*; *Cuesta et al., 2020*). These changes also disrupt primary production (*Gaglio et al., 2017*; *Rey-Romero, Domínguez & Oviedo-Ocaña, 2022*), thereby influencing the survival of some bird species that are directly dependent on resources such as invertebrates (*Tirozzi et al., 2021*). Climate changes pose a particular threat to bird species with small populations that are restricted to high-altitude areas (*Naveda-Rodríguez et al., 2016*; *Restrepo-Cardona et al., 2022*).

The Andean Ibis (*Theristicus branickii*) is a bird species present in high Andean habitats such as the paramo and the high Andean grassland ecosystems above 3,000 m a.s.l. (*Restall, Rodner & Lentino, 2007*; *Schulenberg et al., 2007*; *Collar & Bird, 2011*). The estimated habitat of the Andean Ibis spans approximately 1,080,000 km² (2023) from northern Chile, Bolivia, Peru and Ecuador (*Fjeldså& Krabbe, 1990*; *Clements, 2019*). The Andean Ibis is known to have particular adaptation and survival strategies (*Bakkeren et al., 2020*; *Dawson et al., 2020*) with specific habitat requirements such as proximity to wetlands in high-humidity areas for feeding, and water bodies adjacent to rocky sites for nesting (*Vizcarra, 2009*; *Collar & Bird, 2011*; *Alcocer, 2014*; *West, 2014*; *Ennis et al., 2019*; *Naveda-Rodríguez et al., 2020*; *Luzuriaga-Neira et al., 2021*; *Muñoz et al., 2021*). The species is categorized as Near Threatened (NT) by the International Union for Conservation of Nature and there is evidence of a declining population trend (*BirdLife International, 2017*). The Andean Ibis was also considered a subspecies of *Theristicus melanopis* until 2019 (*Remsen Jr et al., 2023*), further contributing to the limited knowledge about its ecology, biology, current distribution and environmental requirements.

Ecological niche models are one of the most widely employed methods in ecological studies for identifying the relationship between habitat presence/availability and environmental variables that influence species presence (*Araújo et al., 2019*; *Zurell et al.,*

PeerJ ________________________________________________

*2020*). These models facilitate the construction of geographic distribution maps, providing current and future projections (*Soberon, 2011*). Various statistical methods can be applied to develop ecological models using presence, absence, or pseudo-absence records. The information derived from the analysis and the projection of localities or habitat suitability onto maps plays a crucial role in informing stakeholder decisions and conservation efforts.

This study aims to develop ecological niche models of the Andean Ibis throughout its identified geographic range and cover the following: (1) identification of the environmental variables which significantly influence the presence of the species; (2) generation of potential distribution maps based on the developed model; and (3) identification of distribution overlap with protected areas and the existing human threats impacting the potential species distribution. The results of this study enhance the understanding of the ecological requirements of the Andean Ibis and its current species distribution. This study also identifies potential protection areas and human threats to the Andean Ibis that can aid conservation initiatives and decision-making processes.

## MATERIAL AND METHODS

### Study area

The scope of the study was restricted to the countries where the presence of the Andean Ibis has been reported (*Schulenberg et al., 2007*; *BirdLife International, 2017*; *Freile & Restall, 2018*; *Clements, 2019*) and the Andean region where previous investigations on the ecology and biology of the species have been conducted (*Vizcarra, 2009*; *Collar & Bird, 2011*; *Alcocer, 2014*; *West, 2014*; *Ennis et al., 2019*; *Naveda-Rodríguez et al., 2020*; *Luzuriaga-Neira et al., 2021*; *Muñoz et al., 2021*; see Table S1).

### Occurrences and explanatory variables

Presence records of *T. melanopis branickii* and *T. branickii* from 2003 to 2020 were obtained from the Global Biodiversity Information Facility (GBIF; https://gbif.org; *GBIF, 2019*), a platform that collects e-Bird registration data (https://ebird.org/home). Unreliable records, including those from a dubious source, those with sampling bias and duplicate records, were eliminated from the database. To mitigate potential spatial biases that could impact the models (*Boria et al., 2014*), a spatial filter was applied using the SDM toolbox in ArcMap (*Brown, Bennett & French, 2017*). This filtering procedure eliminated occurrences within less than 10 km from their nearest neighbors. Finally, the database was randomly divided into two groups: 70% of the data was separated for model development, while the remaining 30% was reserved for model evaluation (*Huberty, 1996*). A total of 26 environmental variables were identified that represented the ecological requirements of the species (Table 1). The topographic characteristics such as altitude, slope and roughness index were obtained from digital elevation models at 30 m spatial resolution (*Jarvis et al., 2008*). The hydrographic characteristics were obtained from the HydroSHEDS project (*Lehner & Grill, 2013*; *Messager et al., 2016*) at ∼500 m resolution, including the Euclidean distance of lakes (dist_lakes), presence of lakes (pres_lakes) and presence of rivers (pres_river) variables. The bioclimatic conditions were provided by Wordclim version 2.1 and represented 19 variables with a resolution of 1 km (*Fick & Hijmans, 2017*). Vegetation
**Table 1** Environmental and climate variables used to the distribution models for Andean ibis (*Theristicus branickii*) in South America.

| Enviromental variable | Description | Source |
|---|---|---|
| *altitude* | Mean elevation: meters above the sea level (m a s l.) | SGIAR-CSI SRTM 90m (*Jarvis et al., 2008*) |
| *slope* | Average slope in percentage | SGIAR-CSI SRTM 90m (*Jarvis et al., 2008*) |
| *roughness* | Roughness index. Field variability | SGIAR-CSI SRTM 90m (*Jarvis et al., 2008*) |
| *dist_lakes* | Lakes distance in meters using Euclidean distance | Hydrolake (*Messager et al., 2016*) |
| *pres_lakes* | Lakes presence | Hydrolake (*Messager et al., 2016*) |
| *pres_river* | River presence | Hydrorivers (*Lehner & Grill, 2013*) |
| *NDVI* | Normalized Difference Vegetation Index. Average values 2000-2019 | MODIS (*Didan, 2015*) |
| *BIO1* | Annual Mean Temperature in °C | Bio-climate variables WorldClim (*Fick & Hijmans, 2017*) |
| *BIO2* | Mean Diurnal Range (Mean of monthly (max temp - min temp)) in °C | Bio-climate variables WorldClim (*Fick & Hijmans, 2017*) |
| *BIO3* | Isothermality (BIO2/BIO7) (×100) in °C | Bio-climate variables WorldClim (*Fick & Hijmans, 2017*) |
| *BIO4* | Temperature Seasonality in °C (standard deviation ×100) | Bio-climate variables WorldClim (*Fick & Hijmans, 2017*) |
| *BIO5* | Max Temperature of Warmest Month in °C | Bio-climate variables WorldClim (*Fick & Hijmans, 2017*) |
| *BIO6* | Min Temperature of Coldest Month in °C | Bio-climate variables WorldClim (*Fick & Hijmans, 2017*) |
| *BIO7* | Temperature Annual Range (BIO5-BIO6) in °C | Bio-climate variables WorldClim (*Fick & Hijmans, 2017*) |
| *BIO8* | Mean Temperature of Wettest Quarter in °C | Bio-climate variables WorldClim (*Fick & Hijmans, 2017*) |
| *BIO9* | Mean Temperature of Driest Quarter in °C | Bio-climate variables WorldClim (*Fick & Hijmans, 2017*) |
| *BIO10* | Mean Temperature of Warmest Quarter in °C | Bio-climate variables WorldClim (*Fick & Hijmans, 2017*) |
| *BIO11* | Mean Temperature of Coldest Quarter in °C | Bio-climate variables WorldClim (*Fick & Hijmans, 2017*) |
| *BIO12* | Annual Precipitation in mm | Bio-climate variables WorldClim (*Fick & Hijmans, 2017*) |
| *BIO13* | Precipitation of Wettest Month in mm | Bio-climate variables WorldClim (*Fick & Hijmans, 2017*) |
| *BIO14* | Precipitation of Driest Month in mm | Bio-climate variables WorldClim (*Fick & Hijmans, 2017*) |
| *BIO15* | Precipitation Seasonality (Coefficient of Variation) in mm | Bio-climate variables WorldClim (*Fick & Hijmans, 2017*) |
| *BIO16* | Precipitation of Wettest Quarter in mm | Bio-climate variables WorldClim (*Fick & Hijmans, 2017*) |
| *BIO17* | Precipitation of Driest Quarter in mm | Bio-climate variables WorldClim (*Fick & Hijmans, 2017*) |
| *BIO18* | Precipitation of Warmest Quarter in mm | Bio-climate variables WorldClim (*Fick & Hijmans, 2017*) |
| *BIO19* | Precipitation of Coldest Quarter in mm | Bio-climate variables WorldClim (*Fick & Hijmans, 2017*) |

cover was represented by the average vegetation index (NDVI) for the period 2000–2019, obtained from MODIS Terra Vegetation Indices, whose original resolution is 1 km (*Didan, 2015*). Both the presence records and the environmental variables were edited in ArcGIS 10.5 (Esri, Redlands, CA, USA) to obtain the same spatial projection (WGS84 UTM 19S), model extension (background area) and 1 km spatial resolution.

## Data analysis

The maximum entropy method (MaxEnt, (*Phillips et al., 2006*)) in the R program *dismo* package, following the methods outlined by *Hijmans et al. (2022)*, was used to identify the ecological niche of the Andean Ibis. MaxEnt is widely used in ecological studies (*Merow, Smith & Silander, 2013*) because it produces robust models using a small number of samples compared to other methods (*Phillips et al., 2006*; *Merow, Smith & Silander, 2013*). This method requires presence-only records and creates random samples (known as background points) within the landscape to build ecological niches. Another advantage of

MaxEnt is the possibility to choose the characteristic type of the response curves between the variables and the occurrences as well as regularize the complexity of the models (*Phillips et al., 2006*; *Merow, Smith & Silander, 2013*).

Before model development, the background or calibration area (M component), also called the accessible area for the species (*Merow, Smith & Silander, 2013*), was marked using the Andean Ibis presence database, including the southernmost recorded occurrence of the species in Argentina (*Müller, Braslavsky & Chatellenaz, 2021*). These points were used to create a minimal convex polygon. Within this area, 10,000 background points were randomly generated to characterize the environmental context of the background area. To address potential issues of multicollinearity that could affect model performance (*Zurell et al., 2020*), highly correlated environmental variables were identified based on a Pearson coefficient threshold of >0.8 (*Frost, 2020*; Fig. S1). The altitude variable was not eliminated from the selection due to its significance in the species' biogeography. The remaining highly-correlated variables were selected according to their performance (contribution) during model calibration.

During the exploratory phase, calibration models were developed using various settings in the ENMeval package (*Kass et al., 2021*). Three types of model responses linear, quadratic and product were applied both individually and in combination, along with different regularization coefficient values (0, 0.5, 1, 3, and 5). All preliminary models were compared using the lowest Akaike information criterion for small samples (AICc), because of this model's cost-efficiency (*Burnham & Anderson, 2004*). The final model, which exhibited the best fit, was developed through 10 repetitions to obtain the average of its values. Statistical evaluation was conducted using the area under the curve (AUC) and the Boyce Index (*Zurell et al., 2020*; Fig. S2).

The final model was then projected across the four countries that comprise the Andean Ibis's habitat, with the occurrence probability shown in gradients according to the literature. To determine how much of the Andean Ibis's potential distribution is threatened by human disturbance and how much is protected, the "maximization of the sum of sensitivity and specificity" threshold rule (MaxSSS), which has been shown to have high performance value (*Liu et al., 2005*), was used to classify Andean Ibis occurrence probability in a binary map (presence and absence). The map was then overlayed with the boundaries of the protected areas managed by each respective country to assess the extent of the species' potential distribution that fell within protected areas (*Bingham et al., 2019*). To evaluate the anthropic impact on the potential distribution, the map of the Human Footprint Index (represented by population density, land transformation, accessibility and electrical energy infrastructure) was used on the landscapes (*Sanderson et al., 2002*). Human influence impact (HII) is a 0–64 scale with 0 representing theoretically no human impacts and 64 representing the highest human impact. This layer was then overlayed with the potential distribution map to identify the percentage of human disturbance within each protected area along the geographic range.

**Table 2  Potential distribution model of the Andean Ibis in South America. The model includes ten replicates and the average values that determined the final model.** Evaluation values included are area under the curve with test occurrences (TestAUC), and the Boyce Index (0.93). Mean values are in bold.

| | Model replicates | | | | | | | | | | |
|---|---|---|---|---|---|---|---|---|---|---|---|
| | 1 | 2 | 3 | 4 | 5 | 6 | 7 | 8 | 9 | 10 | Mean |
| **Evaluation** | | | | | | | | | | | |
| Test AUC | 0.91 | 0.91 | 0.91 | 0.93 | 0.948 | 0.94 | 0.92 | 0.91 | 0.91 | 0.91 | **0.91** |
| Boyce Index | 0.92 | 0.92 | 0.95 | 0.93 | 0.95 | 0.94 | 0.93 | 0.98 | 0.87 | 0.92 | **0.93** |
| MAXSSS | 0.34 | 0.28 | 0.33 | 0.28 | 0.35 | 0.28 | 0.25 | 0.30 | 0.32 | 0.28 | **0.30** |
| **Contribution %** | | | | | | | | | | | |
| altitude | 46.6 | 31.1 | 40.0 | 30.5 | 47.3 | 35.6 | 30 | 44.1 | 29.4 | 31.1 | **36.57** |
| dist_lakes | 18.9 | 24.1 | 21.7 | 29.5 | 17.3 | 17.6 | 29.7 | 19.3 | 30.7 | 24.1 | **23.29** |
| bio18 (Precipitation of Warmest Quarter) | 0.8 | 10.4 | 12.1 | 4.9 | 1.5 | 0.6 | 1 | 11 | 3.9 | 10.4 | **5.66** |
| pres_river | 4.5 | 4.6 | 5.3 | 5.4 | 4.6 | 4.6 | 5.6 | 5 | 4 | 4.6 | **4.82** |
| slope | 3.6 | 3.9 | 2.2 | 3.4 | 2.8 | 3.4 | 2.2 | 3.6 | 2.2 | 3.9 | **3.12** |
| bio3(Isothermality) | 7.2 | 11.7 | 5.8 | 17.4 | 8.3 | 28.4 | 17.3 | 7.3 | 18.3 | 11.7 | **13.34** |
| roughness | 2.5 | 1.2 | 2.2 | 1.6 | 2.1 | 1.3 | 5.1 | 1.1 | 3.3 | 1.2 | **2.16** |
| bio13 (Precipitation of Wettest Month) | 14.6 | 8.7 | 8.7 | 4.8 | 13.9 | 6.1 | 5.2 | 7.2 | 5.2 | 8.7 | **8.31** |
| bio17 (Precipitation of Driest Quarter) | 1.4 | 4.1 | 2.2 | 2.6 | 2.4 | 2.3 | 3.8 | 1.5 | 2.9 | 4.1 | **2.73** |

## RESULTS

A total of 2,096 Andean Ibis occurrences were recorded (1,357 for Ecuador, 678 for Peru, 58 for Bolivia, and three for Chile, see supplementary data), covering an altitudinal range of 3,667 to 4,642 m, with an average of 4,108.08 m (SD 169.65 m; Fig. 1). Applying the spatial filter, the occurrences were reduced to 154 (108 for the calibration models and 46 for the evaluation). After exploratory analysis, the best model setting was determined to include only nine variables with a linear and quadratic type of response with a regularization coefficient equal to 1 (Table 2).

The final model showed the contributions made by each of the variables (Table 2), with altitude being the most influential (36.57%) followed by distance to lakes (*dist_lakes*; 23.29%) and isothermality (*bio3*; 13.34%). The response curve showed a heterogeneous relation between the environmental values and the model's prediction. Some variables, such as distance to lakes, slope, precipitation of driest quarter (*bio17*), had a negative association, implying that the probability of the presence of the Andean Ibis decreased as these variables' values increased. Conversely, isothermality (*bio3*) and presence of rivers (*pres_river*) had a positive association, meaning the higher their values, the greater the probability of the presence of the species. On the other hand, altitude, precipitation of the wettest month (*bio13*) and roughness shows bell-shaped response curves all peaking above

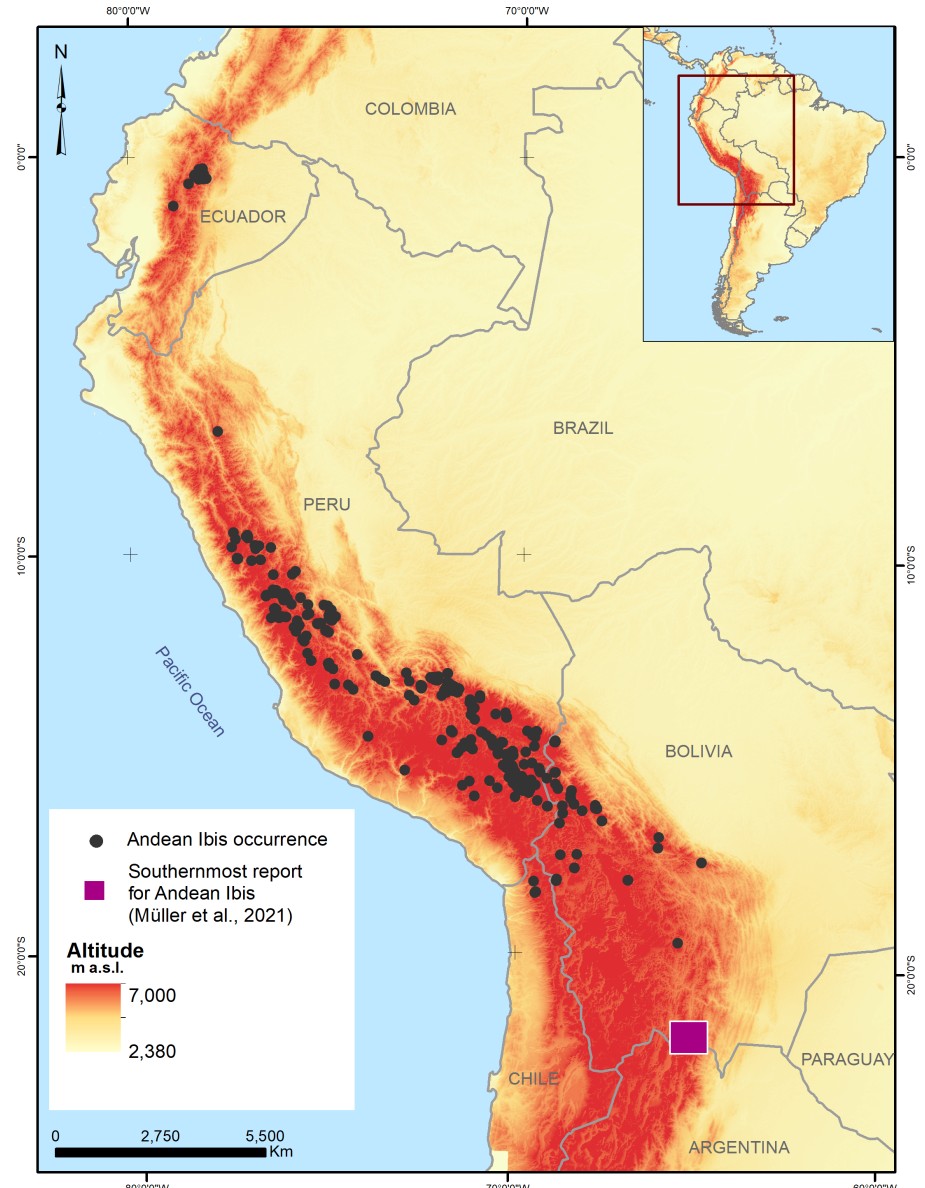

**Figure 1** **Records of the presence of the Andean Ibis (*Theristicus branickii*) in South America between 2003 and 2020.** Data were obtained from GBIF and eBird platforms and specific studies conducted on the species (see references).

0.8 probability of presence. The statistical evaluation of the model yielded satisfactory results (Test AUC 0.914 and Boyce Index 0.93; Table 2).

Figure 2 shows the model projection of the Andean Ibis in the four countries. The colour gradient map shows that the highest occurrence probability values were located principally in Ecuador and Peru, but the distribution area was fragmented. Bolivia shared a distribution area with Peru and also had some independent locations with high occurrence

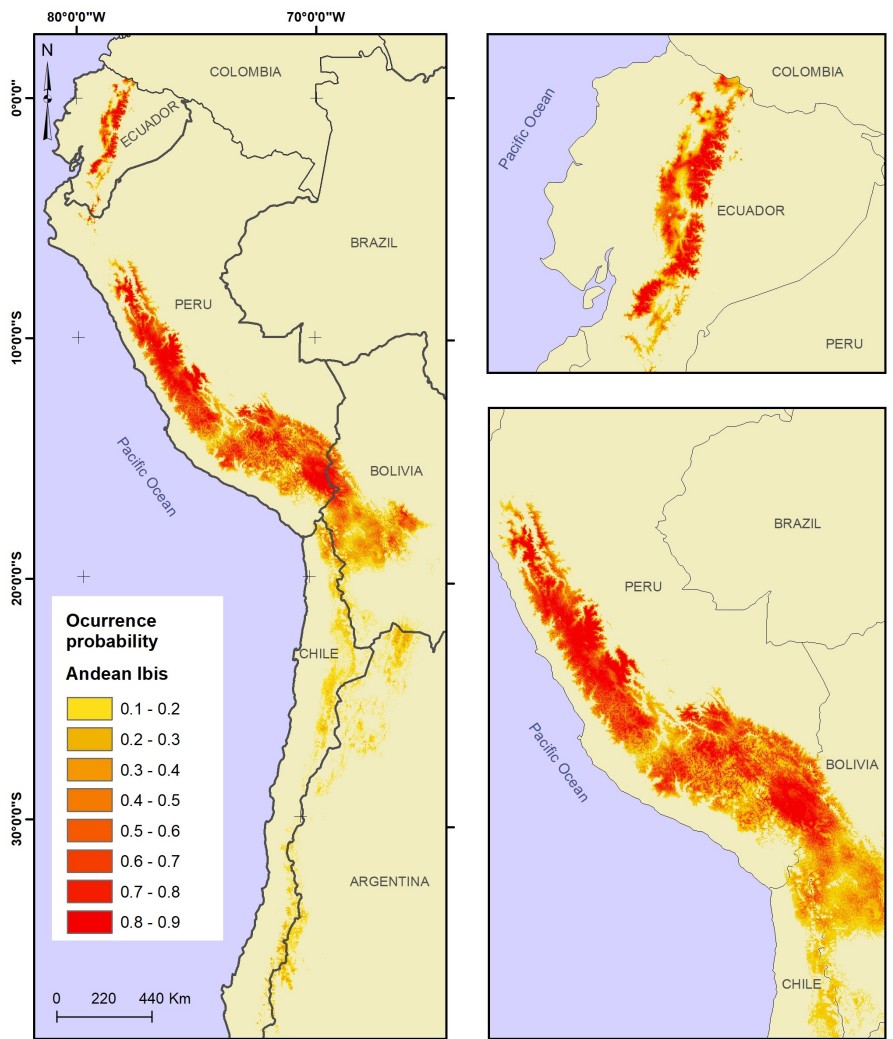

**Figure 2 Potential distribution of the Andean Ibis (*Theristicus branickii*) in South America.** Occurrence probability is shown in gradient color (from yellow to red).

probability values, while Chile showed a very fragmented distribution with the lowest occurrence probability value.

Finally, the binary classification of the prediction was divided with a threshold value equal to 0.30. Figure 3 shows the extent of the potential distribution of the Andean Ibis was 300,095.00 km² at the regional level, with 21,585.85 km² (7.19%) in Ecuador, 216,031.33 km² (71.99%) in Peru, 57,020.64 km² (19.00%) in Bolivia and 5,457.17 km² (1.82%) in Chile (Table 3, Fig. 3). Anthropic disturbance (human footprint) covered ~48% of this potential distribution, while protected areas only covered ~10% (Table 3). Within national territories, the extent of anthropic disturbance was similar in three of the four countries (45% in Ecuador, 47.12% in Peru, and 53.60% in Bolivia), but was much lower

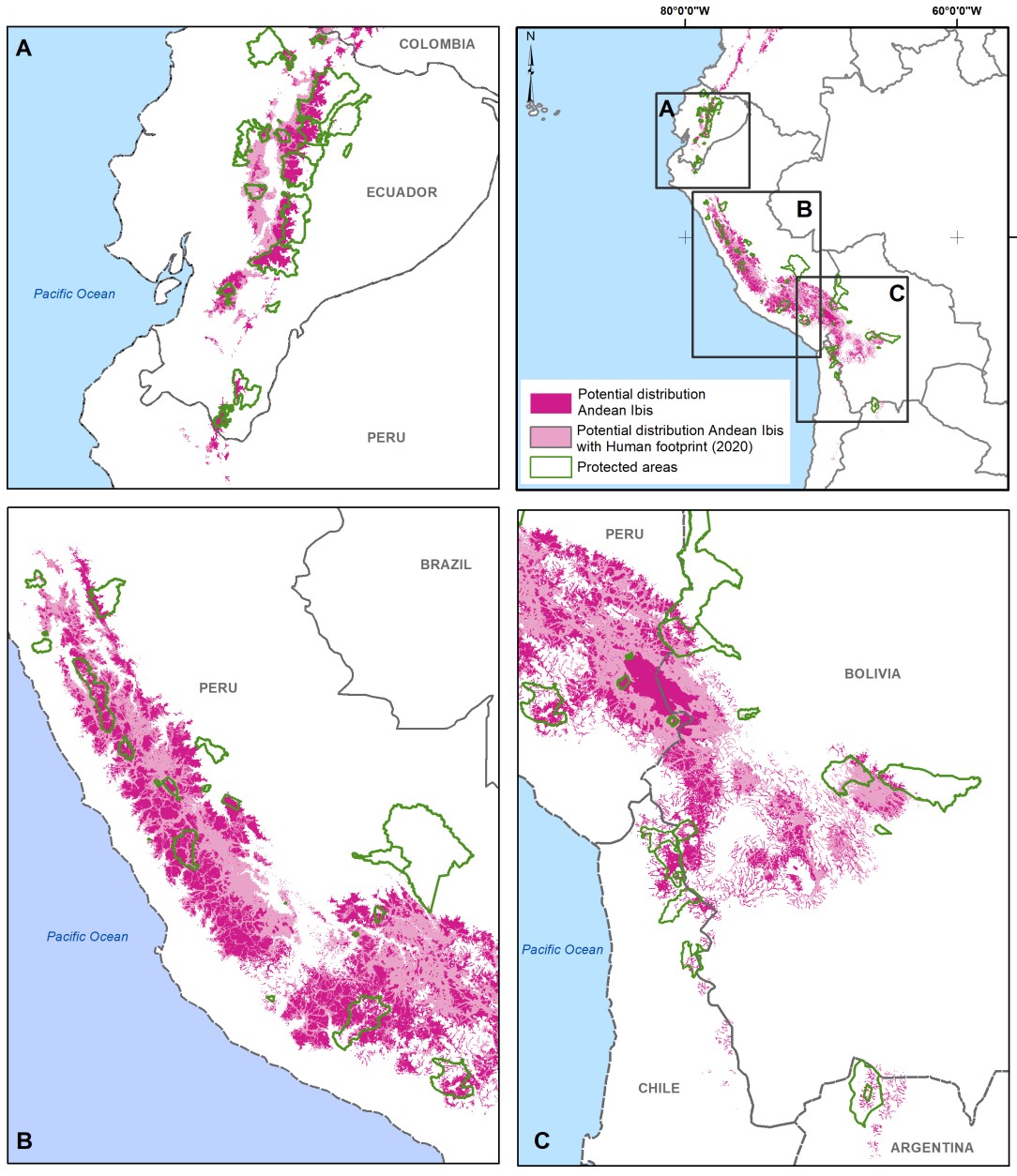

**Figure 3 Potential distribution of the Andean Ibis (pink) in South America.** (A) Ecuador; (B) Peru; (C) Perú, Chile and Bolivia. National protected areas (green borders) and human footprint areas (light pink; WCS, WC.S. 2005) are included. (A) Human footprint, protected areas and potential distribution in the Ecuadorian and north Peruvian zones; (B) human footprint and protected areas in Peruvian zones, and (C) human footprint, potential distribution and protected areas in southern Peruvian, northeast Bolivia, and north Chile zones.

Chile (~7%). Ecuador and Chile had the largest percentage of protected areas within the potential distribution areas (35.20% in Ecuador and 34.30% in Chile; Table S3).

**Table 3 Quantification of the potential distribution of the Andean Ibis (*Theristicus branickii*) in South America.** The geographic range, human disturbed areas, and protected zones are described in the national territories of Ecuador, Peru, Bolivia, and Chile.

| Country | Potential distribution of the Andean Ibis | | | | | |
|---|---|---|---|---|---|---|
| | National territory (km²) | National territory (%) | Anthropogenic disturbance (km²) | Anthropogenic disturbance (%) | Protected areas (km²) | Protected areas (%) |
| Ecuador | 21,585.85 | 7.19 | 9,713.12 | 6.82 | 8,443.10 | 28.62 |
| Peru | 216,031.33 | 71.99 | 101,801.87 | 71.46 | 13,642.40 | 46.25 |
| Bolivia | 57,020.65 | 19.00 | 30,563.63 | 21.45 | 5,540.40 | 18.78 |
| Chile | 5,457.17 | 1.82 | 377.18 | 0.26 | 1,872.00 | 6.35 |
| **TOTAL** | **300,095.00** | **100.00** | **142,455.80** | **100.00** | **29,497.90** | **100.00** |

Notes.
Totals are in bold.

## DISCUSSION

This study developed ecological niche models to identify the environmental conditions that influence the presence of the Andean Ibis, and map its potential distribution in the tropical Andes. The results of this study reinforce the theory that altitude is a key factor in determining the presence of the species. This study also found that around half (48%) of the Andean Ibis's geographic range is threatened by human activities, while only 10% overlaps with protected areas.

A meticulous review of Andean Ibis occurrences was conducted to ensure the accuracy of this study, taking into account the potential confusion with the records of the *Theristicus melanopis* species. It is worth noting that specific locations in Ecuador, such as Cotopaxi National Park and Antisana National Park, are cited as presence zones for *Theristicus melanopis* in the BirdLife datazone (*BirdLife International, 2023*). However, this published information is likely erroneous, as both species were previously classified as a single species by BirdLife International.

Given the significant disparity in altitudinal habitat ranges between *T. melanopis* and *T. branickii*, it was crucial to carefully scrutinize the occurrence records. The review aimed to ensure that only reliable and accurate data were included in the analysis, thereby minimizing any potential misclassification or confusion between the two species.

The limits of the Andean Ibis occurrence distribution area were determined to be: in the north Lat. −0.2981512, Long. −78.1287425 in Ecuador, in the south Lat. −18.4518884, Long. −70.0663054 in Chile, and in the southeast Lat. −19.6002593, Long. −65.7204267 in Bolivia. Unusual occurrence locations were not considered in the model database. One example of this was the Argentina record (*Müller, Braslavsky & Chatellenaz, 2021*) where the species were registered at about 1,562 m a.s.l. (Lat. −22.283333, Long. −62.683333) in a cloud forest habitat, which is not the typical Andean Ibis habitat. Similarly, records from Ita, Peru, were reported at 157 and 36 m a. s. l. (*Vizcarra, 2009*) and in the Azapa Valley in Chile (Lat. −18.533333, Long. −70.150000; (*Estades, Aguirre & Tala, 2004*)). These recorded occurrences are likely accurate and may represent occasional movements to lower elevation areas during the dry season. These specific occurrences were excluded

from this study's database, but were still considered when designing the background area to ensure an appropriately represented environment context.

The results of the niche model highlight that the most influential variable of the Andean Ibis is altitude (Table 2). Using the threshold of 0.3 (MaxSSS), the current potential distribution area of the Andean Ibis had an altitudinal range of 3,300 and 4,800 m, which is a slightly broader range than the distribution range reported in the literature of 3,700–4,500 m a.s.l. (*BirdLife International, 2017*). High altitude locations in the tropical Andes play an important role for the Andean Ibis and it is important to consider their vulnerability to global warming (*Cuesta et al., 2020*). Modifying the landscape configuration in those localities can shrink the habitat of certain specialist species, like the Andean Ibis, compared to generalist species (*Barnagaud et al., 2012*; *Di Cecco & Hurlbert, 2022*).

Other important variables in Andean Ibis occurrence probability were the distance to lakes and the presence of rivers, which an expected result as lakes and rivers are fundamental foraging sites of the species where its presence has been documented (*Muñoz et al., 2021*). Aquatic microhabitats or wetlands are found in high-altitude locations in the Andes and produce invertebrates and macro-invertebrates. Altitude, bodies of water (*e.g.*, wetlands) and the contribution of annual precipitation (>3,000 mm/year for the paramo and >3,500 mm/year for the puna, respectively; *Flores-López et al., 2016*; *Izquierdo et al., 2018*) are the basic components of the Andean Ibis habitat.

The relationship between altitude, bodies of water and isothermal variables in the high-altitude ecosystems of the Andes, as well as their influence on the potential distribution of the Andean Ibis, aligns with previous research findings on the reproductive patterns of the species. Notably, observations conducted in Antisana National Park provide evidence that the species is present in the volcanic region above 3,500 m a.s.l., with areas close to the Antisana River, wetland areas (lakes) and their tributaries serving as preferred roosting and nesting sites (*West, 2014*; *Luzuriaga et al., 2019*). Similar occurrence and reproductive activity patterns were recorded in Cusco, Peru, at 3,850 m a.s.l., where the species uses the rocky area and the cliffs of the Canepia River to nest (*Alcocer, 2014*).

According to the ecoregion classification of *The Nature Conservancy (2003)*, the potential distribution of the Andean Ibis comprises mainly the paramo highlands in North and Central Andes, the Peruvian Yungas, and the montane dry forest in the south of Bolivia (Fig. 4). Although the statistical evaluation of the model shows it is reliable, the empirical evaluation of the distribution projection indicates that there may be a slight overestimation in some areas, specifically in the southeast of Ecuador (*e.g.*, Podocarpus National Park), where the species has not been observed. This overestimation in models using MaxEnt has been previously reported in rare species, with narrow niches and sampling biases. For example, in the distribution model of the Hose's civet (*Diplogale hosei*) researchers found that the data was biased towards localized areas (easy access) and error-corrected by manipulating the background data (*Kramer-Schadt et al., 2013*). The replicate model shows the potential area, but not the occupied area, of new localities available for the species in the future.

Anthropic activities (human footprint) cover ~48% of the extent of the potential distribution of the Andean Ibis, with the most significant impact in Peru. The impact

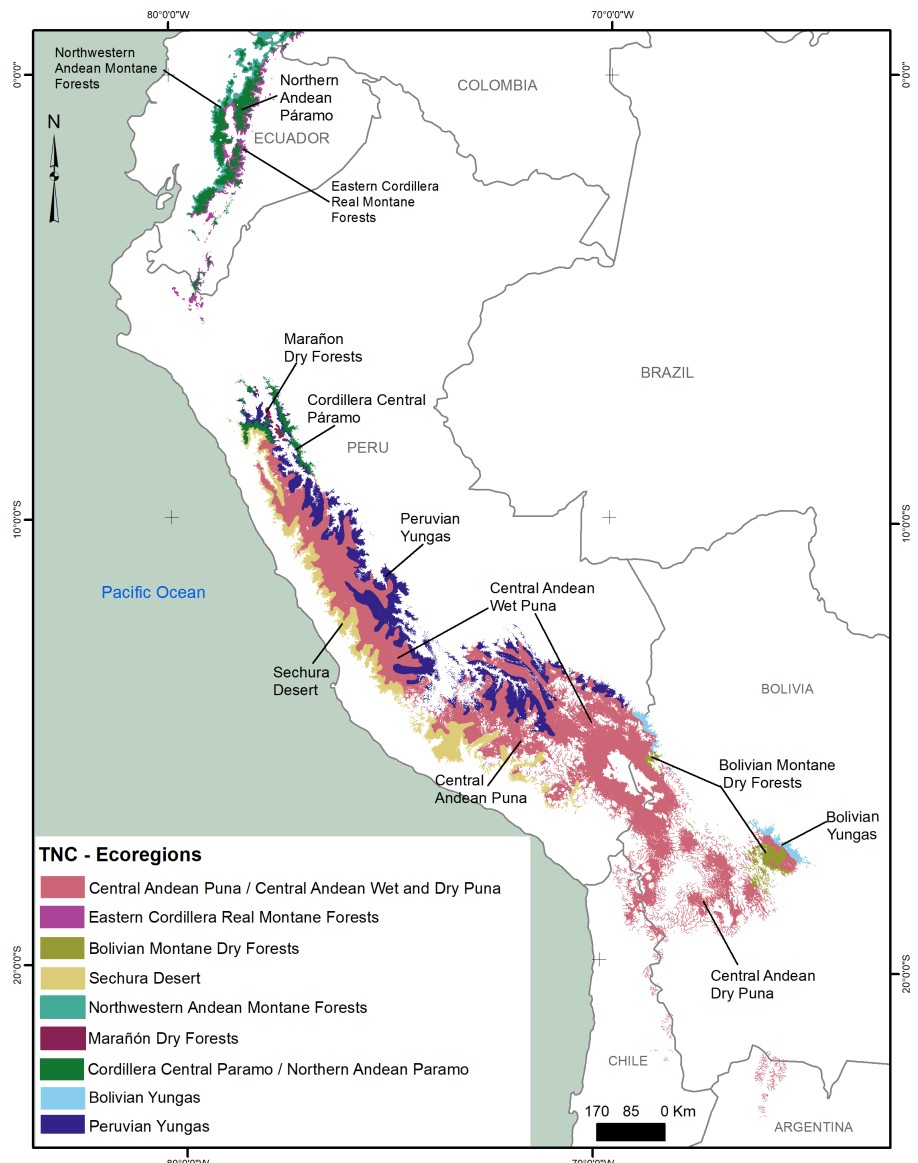

**Figure 4** **South American ecoregions (*The Nature Conservancy, 2003*) and the potential distribution of the Andean Ibis.** Eleven ecoregions are in or near the potential distribution areas of the Andean Ibis.

of anthropogenic pressure on avian species and other functional species such as plants and insects in high Andean ecosystems has been reported in previous studies (*Graham et al., 2009*; *Ocampo-Peñuela et al., 2022*) and is an expected phenomenon considering the high population density and demographic activity of Andean cities, which impose negative impacts on essential resources such as water (*Mulligan, 2009*; *Custodio et al., 2018*). In Ecuador, livestock activities and high-altitude crops such as potatoes, cereals and onions, are leading to habitat transformation of the highlands to farmland at an annual rate of 0.95% (*Gaglio et al., 2017*). Furthermore, scientific evidence shows that this transformation, as well as cultivation processes on a small scale, are associated with a

significant change in water quality indicators (*Rey-Romero, Domínguez & Oviedo-Ocaña, 2022*). Another important factor is climate change, which has a direct impact on altitude zones, influencing the retreat of glaciers (*Francou et al., 2003*) where humidity flows affect the structure and composition of the community of aquatic invertebrates (*Cauvy-Fraunié et al., 2015*), altering the food chain of the species.

## CONCLUSIONS

In an ecological niche model, nine environmental variables influenced the distribution of the Andean Ibis, with altitude and distance to lakes contributing a combined 60%. The altitude range of the species distribution was determined to be 975 m, with a higher occurrence probability in high altitude areas (4,108.08 $\pm$ 169.65 SD m a.s.l.) and a distance to lakes of less than 1 km. Thus, the distribution of the Andean Ibis is restricted to Andean ecosystems with water bodies and wetlands, including habitat patches within 11 South American ecoregions. Areas impacted by the human footprint cover nearly half of the distribution area (48%) and the protected area systems in these countries only overlap with a small portion of the distribution area (10%). Moreover, the species has an important potential presence area. However, human activities could threaten Andean Ibis populations by harming their habitats. These factors should be thoroughly studied, focusing on demographic patterns, habitat use and threat identification, to identify the most effective conservation actions. These actions might include reducing the human footprint, promoting connectivity between Andean Ibis distribution zones or including new protected areas for effective species conservation.

## ACKNOWLEDGEMENTS

Special thanks to the students and volunteer technicians: Karla Mena, Diego Cuichán and Marcelo Cuichán.

### Funding

This work was supported by the Universidad Central del Ecuador (PE26), Fundación de Conservación Jocotoco and Rusell E. Train Education for Nature Program-EFN. The funders had no role in study design, data collection and analysis, decision to publish, or preparation of the manuscript.

### Grant Disclosures

The following grant information was disclosed by the authors:
Universidad Central del Ecuador (PE26).
Fundación de Conservación Jocotoco and Rusell E.

### Competing Interests

Roxana Rojas-VeraPinto, is employed by Isnache Project. Michaël André Jean Moens, was employed by Fundación de Conservación Jocotoco. José León is employed by Fundación de Conservación Jocotoco.

## Author Contributions

- Nivia Luzuriaga-Neira conceived and designed the experiments, performed the experiments, analyzed the data, authored or reviewed drafts of the article, and approved the final draft.
- Keenan Ennis conceived and designed the experiments, performed the experiments, analyzed the data, prepared figures and/or tables, authored or reviewed drafts of the article, and approved the final draft.
- Michaël A.J. Moens conceived and designed the experiments, prepared figures and/or tables, authored or reviewed drafts of the article, and approved the final draft.
- Jose Leon conceived and designed the experiments, prepared figures and/or tables, authored or reviewed drafts of the article, and approved the final draft.
- Nathaly Reyes conceived and designed the experiments, prepared figures and/or tables, authored or reviewed drafts of the article, and approved the final draft.
- Agusto Luzuriaga-Neira conceived and designed the experiments, prepared figures and/or tables, authored or reviewed drafts of the article, english editing, and approved the final draft.
- Jaime R. Rau conceived and designed the experiments, prepared figures and/or tables, authored or reviewed drafts of the article, table and figure editing, and approved the final draft.
- Roxana Rojas-VeraPinto conceived and designed the experiments, performed the experiments, analyzed the data, authored or reviewed drafts of the article, and approved the final draft.

## Data Availability

The data is available at GBIF: https://doi.org/10.15468/dl.emayxe.

## Supplemental Information

Supplemental information for this article can be found online at http://dx.doi.org/10.7717/peerj.16533#supplemental-information.

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
