# Peer review of "The Andean Ibis (Theristicus branickii) in South America: potential distribution, presence in protected areas and anthropic threats"

_PeerJ, doi:10.7717/peerj.16533_

## Round 0.1 · original submission · Minor Revisions

Dear Dr. Nivia Luzuriaga-Neira,

The manuscript "The Andean Ibis (Theristicus branickii) in South America: potential distribution, presence in protected areas and anthropic threats" was reviewed by two independent reviewers. While both reviewers found the study interesting, they also found that there is need to improve the clarity of the text in several parts. Both reviewers criticized some figures of the manuscript stating that "Since the areas of habitat suitability were represented with one solid red color, it suggests that these areas are equally suitable for Andrea Ibis." Thus, figures and figure legends must be adjusted accordingly to reviewers' suggestions. I look forward to receive a revised manuscript.

**Language Note:** The review process has identified that the English language must be improved. PeerJ can provide language editing services - please contact us at copyediting@peerj.com for pricing (be sure to provide your manuscript number and title). Alternatively, you should make your own arrangements to improve the language quality and provide details in your response letter. – PeerJ Staff

Reviewer 1 ·

Basic reporting

• While the paper is well written there are some formatting inconsistencies. There are a few instances where the I in Ibis is not capitalized, examples of this are in line 68 as well as in the caption of Table 1 and in the legends of the figures. There are also missing periods at the end of the Figure and Table captions.
• I suggest adding what the climate variables such as bio3, bio17, and bio18 stand for. This will make it easier for the reader to understand the significance of those variables and how they contribute to the model.
• In the Abstract on line 34 the 2 after km should be in superscript.
• There also appears to be extra spaces between the “extreme environmental” on line 53.
• In line 56 I recommend changing the wording slightly. Instead of “...for some taxons, there are still remains of limited information...” to “...for some taxons, there remains limited information...”.
• On line 150 the word remain should be remaining
• On line 244 and 255 there appears to be missing a paratheses after the word respectively.
• In the legend of Figure 1 Altitude is misspelled.

Experimental design

• The methods are well defined in the paper. The authors explain where they pulled their information for the models from in detail and each step in how their models were built.

Validity of the findings

• The one criticism I have about the findings is mostly with the figures. Since the areas of habitat suitability were represented with one solid red color, it suggests that these areas are equally suitable for Andrea Ibis. While lines 161 – 164 explain that a binary threshold was used, the reason that this particular threshold was chosen, as opposed to other thresholds was not justified. But more importantly, thresholding neglects the fact that there will be areas that are more suitable than others. I would suggest using a color gradient to show the various habitat suitability in the area, in addition to a thresholded map. But the choice of threshold needs to be justified regardless.
• The conclusions were well stated and supported by their findings as well as by their sources.

Additional comments

• I would be careful using red and blue so closely in the figures, specifically Figure 2, as that can be hard for colorblind people to distinguish between.

·

Basic reporting

This is an interesting study that focuses on the use of a species distribution model to relate species occurrence of the Andean Ibis (Theristicus branickii) to environmental variables and check the overlap of the potential habitat suitability with protected areas and human footprint. The authors use the popular maximum entropy model (MaxEnt) to derive a map of potential distribution of the Andean Ibis, a species of conservation concern, assessed as Near Threatened (NT) by IUCN.
The manuscript is well-written, but there is a need to improve the clarity of the text for an international readership by improving the English language used. Specific instances that need improvement are found in (not an exhautive list):
1. Abstract: (line 34) The species' distribution compresses…
2. Introduction: (line 95) The results of this research pretend to enhance…
3. Discussion: the bodies of water…
where the current wording makes it difficult to understand. I recommend that you have your manuscript reviewed by a colleague who is familiar with both the English language and the subject matter, or alternatively, hire a professional editing service. Please also change the Spanish names (Andean Bandurria, La Civeta de las Palmeras) to their English equivalents.

Experimental design

No comments

Validity of the findings

Data analysis
Collinearity: Please provide rationale or references why Pearson coefficient threshold of >0.8 was used (why not |r| > 0.75, for example?). Other methods (PCA, VIF values) could be used.

Additional comments

Figures: The legend of Fig. 2 states that human footprint areas are depicted in red.
Fig. 3: It is difficult to separate the various ecoregions.
Other minors comments :
Abstract: (line 29) five countries? Records from Argentina were discarded…
Line 244: >3,000 mm/year paramo. Add "for the" before paramo.
All in all, congratulations to the authors which should improve their manuscript before publication.

---

## Round 0.2 · Minor Revisions

I have compared the revised manuscript with the original submitted version and I believe authors followed reviewers' s suggestions and greatly improved the manuscript. However, I found some inconsistencies between the rebuttal letter and the revised manuscript as follows:

Figure 2: The rebuttal letter has the legend "Fig. 2 Potential distribution of the Andean Ibis (Theristicus branickii) in South America. Occurrence probability is shown in gradient color (from yellow to red)." but the revised manuscript has the legend "Potential distribution of the Andean Ibis (red spots) in South America. A= Ecuador, B=Peru, C= Perú, Chile and Bolivia. National protected areas (green borders) and human footprint areas (red areas) (WCS, WC.S. 2005) are included."

Figure 3: The rebuttal letter has de legend "Figure 3: Potential distribution of the Andean Ibis (Theristicus branickii), protected areas and human threats. Using the MaxSSS threshold rule (0.3) identified as the potential distribution (mustard areas) for the species. Human footprint (gray areas with slight transparency) and protected areas (green borders with diagonal lines) overlap the potential distribution." but the revised manuscript has the legend "South American ecoregions (The Nature Conservancy 2003) and the potential distribution of the Andean Ibis. Potential distribution area is red shadow label. Eleven habitat patch is into or around potential distribution specie’s zone."

For Figure 3 incorporate your "Notes: A: Human footprint, protected areas and potential distribution in the Ecuadorian and north Peruvian zones; B: Human footprint and protected areas in Peruvian zones, and C: Human footprint, potential distribution and protected areas in southern Peruvian, northeast Bolivia, and north Chile zones." into Figures Legend.

Figure 4: The rebuttal letter has the legend "Figure 4: Potential distribution (Theristicus branickii) and Ecoregions proposed by The Nature Conservancy (2003). Only ecoregions that overlap with the distribution are shown being the most important ones Paramo (green border), Peruvian Yungas (purple dots), Puna (diagonal lines) and Bolivian Mountain Dry Forest (light blue net)." and the revised manuscript has the legend "South American ecoregions (The Nature Conservancy 2003) and the potential distribution of the Andean Ibis. Potential distribution area is labeled with red shadowing. Eleven ecoregions are in or near the potential distribution areas of the Andean Ibis."

Please be careful while revising your revised figure legends to avoid any inconsistencies. I would suggest double-checking table legends as well.

---

## Round 0.3 · accepted · Accept

The authors have addressed all of the original reviewers' comments as well as the editorial corrections related to inconsistencies between the first rebuttal letter and figure legends. I believe the manuscript will make a good contribution to PeerJ.